# *Pseudomonas* Flagella: Generalities and Specificities

**DOI:** 10.3390/ijms22073337

**Published:** 2021-03-24

**Authors:** Mathilde Bouteiller, Charly Dupont, Yvann Bourigault, Xavier Latour, Corinne Barbey, Yoan Konto-Ghiorghi, Annabelle Merieau

**Affiliations:** 1LMSM, Laboratoire de Microbiologie Signaux et Microenvironnement, EA 4312, Normandy University, Université de Rouen, 27000 Evreux, France; mathilde.bouteiller7@univ-rouen.fr (M.B.); charly.dupont7@univ-rouen.fr (C.D.); yvann.bourigault@univ-rouen.fr (Y.B.); xavier.latour@univ-rouen.fr (X.L.); corinne.barbey@univ-rouen.fr (C.B.); yoan.konto-ghiorghi@univ-rouen.fr (Y.K.-G.); 2SFR NORVEGE, Structure Fédérative de Recherche Normandie Végétale, FED 4277, 76821 Mont-Saint-Aignan, France

**Keywords:** flagella, *Pseudomonas*, T6SS, flagellar crosstalk

## Abstract

Flagella-driven motility is an important trait for bacterial colonization and virulence. Flagella rotate and propel bacteria in liquid or semi-liquid media to ensure such bacterial fitness. Bacterial flagella are composed of three parts: a membrane complex, a flexible-hook, and a flagellin filament. The most widely studied models in terms of the flagellar apparatus are *E. coli* and *Salmonella*. However, there are many differences between these enteric bacteria and the bacteria of the *Pseudomonas* genus. Enteric bacteria possess peritrichous flagella, in contrast to Pseudomonads, which possess polar flagella. In addition, flagellar gene expression in *Pseudomonas* is under a four-tiered regulatory circuit, whereas enteric bacteria express flagellar genes in a three-step manner. Here, we use knowledge of *E. coli* and *Salmonella* flagella to describe the general properties of flagella and then focus on the specificities of *Pseudomonas* flagella. After a description of flagellar structure, which is highly conserved among Gram-negative bacteria, we focus on the steps of flagellar assembly that differ between enteric and polar-flagellated bacteria. In addition, we summarize generalities concerning the fuel used for the production and rotation of the flagellar macromolecular complex. The last part summarizes known regulatory pathways and potential links with the type-six secretion system (T6SS).

## 1. Introduction

Flagella are found in many organisms, including Gram-negative and Gram-positive bacteria, *Archae*, and eukaryotic cells. This widespread organelle is present in more than 80% of known bacterial species [1]. To date, no homology has been found between this bacterial appendage and that of eukaryotes or *Archae* [2]. Flagella are long, helical, rotatable appendages located on the bacterial cell surface and enable the bacterium to swim in liquid media and swarm upon semi-solid surfaces. In Gram-negative bacteria, this apparatus spans the inner and the outer membranes.

This complex nanomachine allows bacteria to move toward more favourable conditions or to escape a deleterious environment. Such flagellar-driven motility and chemotaxis allow bacteria to colonize many diverse environments. In *E. coli*, flagellar biosynthesis and function use approximately 2% of the total cellular energy [3] and are coordinated with other cellular process, such as biofilm formation, virulence, and physiological state. Many reports on the flagellar apparatus have highlighted a link between flagellar-driven motility and biofilm formation and its dispersion. The adhesion step is crucial for biofilm formation and involves the flagellin protein FliC, the flagellar cap protein FliD, and flagellar movement [4,5]. For example, in the *Pseudomonas aeruginosa* PAK strain, the FliD protein of flagella is essential for mucin adhesion. Bacteria save energy during other biofilm steps by downregulating the expression of this appendage to conserve energy and remain adherent. Thus, the flagellum may also directly act as a sensor. For example, it can detect surface proximity by higher load or slowed rotation, which can then lead to a switch in the bacterial lifestyle towards a sessile, biofilm-like mode [6,7]. Furthermore, the flagellar stator can also be considered as a sensor. Indeed, the alteration of rotation in a highly viscous environment induces differentiation and hyper-flagellation [8].

This extracellular appendage is required for virulence in many bacteria [4]. In *Salmonella enterica* serovar Typhimurium, flagellin monomers are recognized by the immune system by the basolateral epithelial Toll-like receptor 5 (TLR5), which then activates interleukin 8 (IL-8) secretion. These flagellin monomers also provoke neutrophil attraction by activation of the NF-kB pathway [9]. Furthermore, the flagellin filament and chemotaxis are required for the induction of *S. enterica* colitis in streptomycin pre-treated mice [10]. In other microorganisms, such as *P. aeruginosa*, the flagellar appendage plays a role in virulence by favouring internalization of the pathogen by corneal epithelial cells [11]. Finally, Taguchi et al. demonstrated the involvement of flagella in the virulence of the phytopathogen *P. syringae* pv. *tomato* by its inducing the hypersensitive cell-death response [12].

Within Gram-negative bacteria, flagella show many differences in terms of their gene names, localization, number, function, expression, and regulatory pathways. Such disparity explains the difficulties in summarizing knowledge on the flagellar apparatus. Among Gram-negative bacteria, *Salmonella* and *E. coli* species are the most widely studied models for flagella. These bacteria possess flagella (from 5 to 10) located randomly around the cell, thus called peritrichous flagella [13]. On the contrary, the flagella in *Pseudomonas* are located at the pole of the cell and are thus called polar flagella [14]. In most cases, the polar flagellum is unique [15,16], but certain *Pseudomonas* species can possess as many as seven polar flagella [17]. In this review, we focus on the macromolecular machinery of this structure in *Pseudomonas* species, which have been less extensively studied than the well-described flagellar systems of *Salmonella* and *E. coli*. We first describe the flagellar structure, which is well conserved between peritrichous and polar flagellated bacteria. Then, we detail flagellar assembly and highlight the specificity of this process in *Salmonella*/*E. coli* and *Pseudomonas*. The third part of this review explains flagellar fueling. We conclude by summarizing the regulation of *Pseudomonas* flagella, both their production and function, by extrinsic and intrinsic parameters.

## 2. Flagellar Structure

The flagella of Gram-negative bacteria are composed of three elements: the membrane complex, the hook, and the flagellin filament [18]. The membrane-complex anchors the flagella in the cell surface and the hook serves as a flexible linker between the membrane complex and the rigid flagellin filament. Rotation of the filament is essential to propel bacteria in liquid media, resulting in bacterial “swimming” motility. The general structure of flagella is highly conserved among Gram-negative species [8,19], but there are differences, for example, in the flagellin filament, motor proteins, ions used for motive force, and gene names. We summarize the gene names and their functions in *Salmonella, E. coli*, and *Pseudomonas* in Table 1 to avoid any confusion. All genes listed in this table are related to transcription classes based on previous studies [13,20,21].

The membrane complex of the flagellar apparatus is composed of the basal body, which contains four rings, the flagellar-type III secretion system, a stator complex, and a hollow rod, from which are exported other flagellar components, such as “hook” subunits, or flagellin monomers (Figure 1).

The basal body (represented in red in Figure 1) is composed of the C, MS, P, and L rings localized at the different layers of the cell: cytoplasm, inner membrane, peptidoglycan, and outer membrane, respectively.

The MS-ring (membrane–supra-membrane ring) contains approximately 26 copies of FliF protein subunits and is attached to the inner membrane. The P-ring, consisting of FlgI protein, is connected to the peptidoglycan layer and the L-ring, comprised of FlgH subunits, and anchors this macromolecular structure to the outer membrane.

These three “ring-like structures” are connected each other by a rod, which is described later. The C-ring is cytoplasmic and composed of three proteins: approximately 25 copies of FliG, 34 subunits of FliM, and approximately 110 copies of FliN [8,18]. FliM proteins from the C-Ring interact with the chemosensory system (described later) to switch the direction of rotation of the flagella. The FliN proteins provide a binding site for chaperone/substrate complexes before their secretion. The amino-terminal domain of the FliG C-ring protein is essential for interacting with the MS-ring and the two other domains are involved in the switching of rotation [8]. The C-ring is considered to be the rotor of the flagellar motor and is coupled to a stator complex formed by the MotA and MotB proteins (represented in pink in Figure 1). The stator complex is composed of four MotA and two MotB proteins and is essential for producing the ion-motive force [22]. The MotB protein contains a short N-terminal cytoplasmic segment, one transmembrane helix, and a large carboxy-terminal periplasmic domain [23]. This protein is anchored to the peptidoglycan layer through an OmpA-like motif in its carboxy terminal periplasmic domain [8,22]. In addition, the MotA subunits contain four transmembrane alpha-helical segments, two short loops in the periplasm, and two long segments in the cytoplasm that are in contact with the carboxy-terminus of the FliG C-ring proteins [23].

Connected to the basal body, the flagellar-type III secretion system (fT3SS) (represented in orange in Figure 1) corresponds to the export apparatus for the flagellar proteins and shares structural similarities with virulence-related T3SS (vT3SS) [6]. This secretion system is composed of a stable inner-membrane module (containing FliO, P, Q, R, FlhA, and FlhB proteins), called the export gate, and a dynamic cytoplasmic module, called the ATPase complex (containing FliI, FliH, and FliJ proteins). The FliOPQR complex of the export gate constitutes a central channel with a diameter of 1.5 nm. FlhA and FlhB are the most highly studied proteins of the export gate structure, as they play a key role in the switching of substrate specificity for export [24], which governs the secretion of junction proteins by fT3SS instead of hook monomers. Upon the switch of substrate specificity, the affinity of fT3SS for hook monomers decreases in favour of a higher affinity for junction proteins, resulting in an increase in their secretion. The FlhB protein contains two domains that are linked by a flexible linker. The first domain is composed of four transmembrane α-helices and the second, the cytoplasmic domain. The cytoplasmic domain of FlhB contains a conserved NPTH (asparagine/proline/threonine/histidine) motif that is suppressed by autocleavage, which allows the switch of substrate specificity of the export channel [25]. The protein FlhA is assembled into a homononameric ring that is believed to change its conformation, thus supporting the switch in substrate specificity [24]. Moreover, FlhA has ion channel activity, and its interaction with FliJ of the ATPase complex allows utilization of the proton motive force (PMF) for protein export [26]. Concerning the ATPase complex, FliI is a Walker-type ATPase homologous to α and β subunits of the F_1_ ATPase [27]. This protein assembles to form a hexameric structure (ATPase) at the base of the flagellum and associates with FliJ monomers to form the FliI_6_/FliJ complex, responsible for ATP hydrolysis [26]. Furthermore, FliI is associated with another protein, FliH, in the FliH_2_/FliI complex. The FliH protein interacts with FliN and FlhA proteins from the C-ring and export gate, respectively. FliH is considered to be a negative regulator of the ATPase and binds to the amino-terminal domain of FliI [26]. In addition, FliJ appears to recycle minor chaperone proteins. It is important to note that the ATPase complex is able to interact with C-ring and export gate components, but these interactions are dynamic to enhance ATPase functionality [28].

The rod (represented in yellow in Figure 1) is a hollow structure that is attached to the basal body and the export gate by FliE, considered to be a rod adaptor [29]. Approximately nine monomers of FliE are assembled and are assumed to connect the MS-ring to the proximal rod, via FlgB [8]. The rod structure is approximately 8 to 14 nm in diameter and can be divided into the proximal and distal rod [19]. The proximal rod contains approximately six subunits of FlgB, FlgC, and FlgF. The distal rod is formed by two ring-like layers of FlgG (approximately 26 monomers) and serves as a connector between the proximal rod and the hook [13].

The hook (represented in green in Figure 1) is composed of 11 circularly arranged protofilaments of the FlgE protein [23] and approximately 120 monomers are essential to assemble a structure of approximately 55 nm length in *Salmonella* [18]. The specific FlgE extensive interactions explain why each of the 11 protofilaments can be compressed or elongated relatively freely to allow the flexible bending of the entire hook structure [30]. FlgE proteins are structurally similar to the FlgG proteins of the distal rod, which may explain the direct interaction between these two components [13].

The long flagellin filament (in shades of blue Figure 1) is attached to the hook-basal-body complex (HBB complex) by FlgK and FlgL (also named HAP1 and HAP3 for hook-associated protein, respectively), which are considered to be junction proteins (in white in Figure 1). The flagellar filament is approximately 10 µm in length and contains approximately 20,000 subunits of flagellin (FliC) monomers. Despite differences in the flagellin filament between Gram-negative bacteria, it is generally comprised of 11 protofilaments of flagellin subunits [2]. The filament switches between left- and right-handed supercoiled forms when the bacteria switch their swimming mode. Supercoiling is produced by two different packing interactions of flagellin called L and R [31]. The “short” state results from a protofilament having a right-handed inclination (the R-state), whereas the “long” state results from a protofilament having a left-handed inclination (the L-state) [2]. The change between L-handed and R-handed supercoiled flagellin leads to a decrease in the inter-subunit distance of 0.8 Å [31]. In *Salmonella enterica* serovar Typhimurium, a flagellin subunit is approximately 500 amino acids [32] and is composed of four linear domains (D_0_ to D_3_ domains) [33]. The D_0_ and D_1_ domains are well conserved and allow the core interaction. The D_0_ domain is located in the inner tube of the filament and D_1_ appears to be responsible for the switch between R-handed/L-handed assembly [31]. Thus, these two domains are essential to assemble the protofilament and are connected to other flagellin monomers. The D_2_/D_3_ domains are hypervariable, protrude outward from the core, and are considered to be antigenic domains [2,34].

Finally, FliD is the protein of the flagellin cap (represented in purple in Figure 1) and is essential for assembly of the flagellar filament. It is the only cap protein that remains attached until proper assembly of the flagellum. The flagellin cap is composed of a pentamer of FliD protein, creating a pentagonal plate structure of “five-leg domains”. A cavity in this filament cap permits the crossing of a subunit of flagellin and appears to act as a folding chamber. The 40 N-terminal amino acids are conserved, and the 50 C-terminal amino acids correspond to the “leg domains”.

Similar to the flagellin amino-acid sequence and the supercoiled form, the diameter of the inner tube can differ between species. The internal diameter in *Salmonella* is approximately 25 Å and the total diameter 230 Å, whereas the corresponding values in *Pseudomonas* species are 25 Å and 170 Å, respectively [2]. In *P. aeruginosa*, proteins resulting from *fliC* expression are classified into two categories, a-type and b-type flagellin, based on their antigenic properties and size [35]. The b-type flagellins are homogenous, with conserved sequences, in contrast to the a-type flagellins, which are more heterogenous and smaller in size. The carboxy- and amino-terminal regions of the two types are identical. The a-type flagellin is found in the *P. aeruginosa* PAK strain, whereas the PA01 strain has b-type flagellin, resulting in different antigenic responses [36]. Flagellin can be post-translationally modified, and such modifications play a key role in various processes. For example, in *S. enterica* serovar Typhimurium, flagellin monomer methylation enhances its adhesion to hydrophobic host-cell surfaces and contributes to efficient gut colonization and host infection [37]. Furthermore, glycosylation of flagellin is important for the virulence of *Pseudomonas*, such as that of *P. syringae* [12,38] or *P. aeruginosa* [39]. Unlike other bacteria that possess glycosylated flagellin, this post-translational modification does not influence flagellar stability in *Pseudomonas* [40]. In these bacteria, flagellin glycosylation modifies its hydrophobic properties, influencing the interaction between the flagella and various surfaces [41].

## 3. Dynamics of Flagellar Assembly

The production of functional flagella involves more than 50 genes (components of flagella, chaperones, regulators) [1] and a large amount of energy is thus required, implying tight regulation. Flagellar genes are expressed in a hierarchical manner, consistent with the assembly process [42]. Moreover, many other regulatory mechanisms are involved to ensure proper assembly of the flagellum, such as post-translational modifications, substrate affinity, mRNA stability, and the control of “hook” length [43]. Briefly, flagellar assembly proceeds sequentially, starting from the cytoplasmic components and finishing with the distal ones. The components of the basal body and rod are assembled first, followed by the hook components, connecting layers (FlgK/L), and flagellin filament. Moreover, proper assembly of the flagella requires the addition of a specific cap to each growing structure. The assembly can be divided into nine arbitrary stages, as depicted in Figure 2.

Concerning assembly of the distal rod, the FlgJ distal rod cap protein must be present at the tip of the distal rod to degrade the peptidoglycan layer [13] and attach the monomers to each other. FlgD, the hook cap protein, is essential for polymerization of the FlgE subunits to the growing hook structure, and FliD must be added prior to FliC polymerization. Here, we first focus on general flagellar assembly; then on the specificities of *Salmonella* and *E. coli*, which are quite well described; and last on *Pseudomonas* flagellar assembly, which is less well documented but relatively similar to that of bacteria of the *Vibrio* genus.

### 3.1. General Scenario of Assembly

The first step of flagellar construction, presented in Figure 2A, corresponds to assembly of the basal-body and proximal rod. First, FliF is inserted into the inner membrane, requiring FliG for efficient MS-ring formation [8]. Then, FlhA subunits from the export gate are assembled into nonameric rings, and other fT3SS components are added to the nascent flagellar structure. After completion of the export gate, FliM/FliN_4_ complexes are assembled through interactions between FliG and FliM to form the C-ring. The ATPase complex is then added to the structure to permit the export of flagellar substrates in a PMF-dependent manner. The rod adaptor FliE self-assembles at the periplasmic surface of the MS-ring to allow proper assembly of the proximal rod proteins FlgB/C and then FlgF, in that order, followed by the distal rod cap protein FlgJ.

The second step (Figure 2B) consists of rod extension. FlgJ acts as a muramidase enzyme to degrade the peptidoglycan layer, permitting insertion of the distal rod into the peptidoglycan layer [19]. In addition, FlgJ is essential for proper assembly of the FlgG subunits. Interestingly, FlgG appears to be controlled by an intrinsic mechanism, still poorly understood, to allow only two stacks of FlgG subunits to assemble in an individual protofilament [18]. Then, FlgI protein of the P-ring is secreted by the Sec-pathway for localization in the periplasm. Another associated protein, called FlgA (not represented in Figure 2), appears to act as a periplasmic chaperone of FlgI. The highly conserved FlgA probably ushers FlgI to the peptidoglycan layer, where the P-ring is formed [44]. When the distal rod reaches the outer membrane, components of the L-ring (FlgH) are secreted by the Sec-pathway in the periplasm and are then subjected to the outer membrane sorting system *Lol* (localization of lipoproteins) [44,45].

The hook cap insertion step (Figure 2C) is preceded by the loss of FlgJ during formation of the P-ring [25]. The hook cap protein FlgD is secreted by fT3SS, at the tip of the nascent structure, to permit polymerization of the hook monomers.

During hook extension (Figure 2D), FlgE hook subunits are secreted by fT3SS. Formation of the hook structure is crucial for flagellar function, explaining why this step is tightly regulated. It is important to note that the genes encoding the hook cap protein and the hook subunits are clustered in the same operon. Lee and Hughes proposed that *flgE* mRNA translation is regulated by a posttranscriptional mechanism to permit the sequential expression of *flgD* and *flgE* [46]. They also showed that the last 15 nucleotides of *flgD* are essential for the binding of ribosomes to the *flgE* ribosome binding site (RBS).

Hook completion occurs when a determinate length of the hook structure is achieved. Such control of the length appears to require three proteins: FlhA, FlhB, and FliK. Numerous studies have demonstrated that the protein FliK is essential, and indeed, *fliK* deletion results in a “polyhook” appendage phenotype. The FliK protein is considered to be a molecular ruler, essential for the assembly of a hook of sufficient length, but its functional activity is not fully understood. Nevertheless, three models have been proposed [18] and FlhB autocleavage activity is required to stop FlgE secretion in all three. The first and most widely accepted model proposes that FliK protein acts as a molecular ruler. During hook assembly, the amino-terminal domain of FliK interacts with FlgD, and the carboxy-terminal domain of FliK interacts with FlhB. This double interaction is responsible for elongation of the FliK protein until its maximal length is reached, which corresponds to hook completion. In the second model, the “C-ring cup model”, 30 subunits of FlgE are stored in the C-ring cavity before their secretion and this step is repeated four times to achieve 120 subunits. The last proposed model is based on the half-life of the proteins, which limits the number of FlgE subunits in the hook. Completion of the hook structure is coordinated with the specificity substrate switch, which allows the export of late flagellar components, including FlgM secretion through fT3SS (Figure 2E). FlgM is the anti–sigma of the sigma 28 factor (FliA). This sigma factor is considered to be flagella-specific, despite its involvement in the regulation of other genes, such as those related to biofilm formation [47]. FliA is expressed during early flagellar class gene expression but is sequestered by FlgM until the hook achieves its correct length. After FliM secretion, the FliA sigma factor is then free in the cytoplasm for the expression of target genes.

After the release of FlgM, hook-associated proteins (FlgK and FlgL) are exported to ensure junction protein insertion (Figure 2F). These proteins are chaperoned by FlgN, which also promotes FlgM translation. However, FlgN is not essential for FlgK/L secretion [43].

Flagellin cap proteins (FliD) are then exported to the tip of the nascent structure and their chaperones FliT are released (Figure 2G). The flagellin subunit FliC and its chaperone FliS are then supported by the fT3SS, where the flagellin FliC is exported after post-translational modifications, ensuring filament extension (Figure 2H). FliS plays a key role in controlling filament length, as *fliS* deletion results in a shorter flagellum [48]. Length control of the filament may vary between species; some species have a filament of defined length, whereas others have continuously growing filaments [6].

Functional flagella require assembly of the stator proteins MotA and MotB (Figure 2I). MotB is inserted into the peptidoglycan layer, probably with another protein that remodels the peptidoglycan [19]. In addition, four MotA subunits are placed around a MotB_2_ homodimer to form the stator complex and allow torque generation. An additional protein, called FliL, appears to interact with the MotAB complex. Although the function of FliL is yet to be clarified, this flagellum-associated protein increases stator engagement and probably acts as an ion availability sensor [22].

It is important to note that the affinities of the chaperone proteins (i.e., FlgN, FliT, and FliS) with the FliH_2_/FliI ATPase complex are different, which defines the order of flagellar protein export [49]. Certain chaperone proteins can also have a regulatory role. Indeed, FlgN (junction protein chaperone) enhances *flgM* expression [13] and FliS (FliC chaperone) is able to bind to FlgM to protect it from proteolysis. Thus, FliS competes with FliA for binding to FlgM [50].

### 3.2. Specificities of Salmonella and E. coli

In peritrichous flagellated species, such as *Salmonella* and *E. coli*, the expression of more than 50 flagellar genes is organized into a transcriptional hierarchy that is based on three promoter classes that are temporally regulated in response to assembly. First, the expression of flagella genes begins in *Salmonella* and *E. coli* with the expression of the *flhDC* operon, containing the class I genes (Figure 3A). The expression of this operon is sigma 70 dependent [13]. The resulting proteins form the transcriptional activator complex FlhD_4_C_2_, which is involved in other regulatory processes. Its activity is dependent on its own stability and the action of the ClpXP protease [13,51]. The transcriptional activator FlhD_4_C_2_ then interacts with sigma 70 to allow expression of the class II genes [13]. The expression of these genes is necessary to produce the structural elements of the membrane complex, hook proteins, filament cap protein (FliD), and regulators (FliA, FlgM, FliZ, and FliT) (Figure 3A). As explained above, hook completion induces the substrate specificity switch and the anti-sigma factor FlgM is secreted through fT3SS. In *E. coli* and *Salmonella*, release of the FliA sigma factor in the cytoplasm triggers the expression of late substrates corresponding to class III genes. During class III promoter activation, the genes encoding proteins of the filament, the motor (*mot*), and chemotaxis system (*che*) are expressed. In addition, the FliA sigma factor is able to activate the transcription of numerous genes already expressed during the expression of class II genes via FlhD_4_C_2_. The expression of genes involved in flagellin production is only FliA dependent to permit a delay in their expression. The proteins encoded by these genes improve the secretion of other earlier components.

Among regulatory proteins, FliZ activates FlhD_4_C_2_-dependent gene expression [52] and also indirectly inhibits FlhD_4_C_2_ degradation by ClpXP [48]. The FliT chaperone protein represses FlhD_4_C_2_-dependent transcription by physically interacting with this master activator to decrease class II gene expression [43].

The *fliL* gene is expressed during class II promoter activation. The resulting protein FliL is found in all bacterial genera, and *fliL* deletion leads to diverse developmental and motility phenotypes, depending on the bacterial genus [53]. In *Salmonella* and *E. coli*, FliL is essential for “swarming motility”, probably by increasing stator engagement and torque generation instead of increasing stator number [22,53].

Flk (also called Fluke protein or RflH to avoid confusion with FliK) is a protein encoded by a gene found far from other flagellar genes and has only been described in *E.coli* and *Salmonella* [54]. This protein appears to act as a second lock to coordinate hook assembly with the substrate specificity switch and prevent premature secretion of FlgM. The single “transmembrane-spanning domain”, located in the carboxy terminus of Flk, anchors Flk to the inner membrane [55]. Karlinsey et al. first proposed that Flk is essential for coupling the translation of *flgM* mRNA and FlgM secretion by fT3SS [56]. “How the Flk protein acts as a second lock” is still unclear. However, studies have provided evidence showing that Flk may serve as a sensor of L- and P-ring formation [54].

### 3.3. Specificities of Pseudomonas

Bacteria of the *Pseudomonas* genus show differences in the number of flagella between different species or even strains. For example, the average number of flagella of *P. syringae* pv. tabaci cells is 2.7, approximately two for the *P. fluorescens* MFE01 strain and only one for both the *P. fluorescens* SBW25 strain and *P. aeruginosa* [57,58]. Other *Pseudomonas* strains may have more flagella, such as *P. putida* PRS2000, which generally possesses five to seven flagella, or *P. fluorescens* C7R12, which can have up to seven [17].

In polar flagellated bacteria, such as *Pseudomonas* or *Vibrio* species, approximately 60 genes are involved in the production of functional flagella [1] and need to be expressed in a hierarchical manner. Due to the polar localization, the assembly process is more sophisticated than in peritrichous bacteria. In *Pseudomonas*, flagellar gene expression is controlled in a four-tiered hierarchy of transcriptional regulation (called classes I to IV) instead of the three classes of enteric bacteria [20] (Figure 3B). Briefly, the class I promoter allows the expression of *fleQ* in a sigma 70-dependent manner. FleQ is considered to be the flagellar master regulator in *Pseudomonas* species and belongs to the enhancer-binding protein (EBP) family. As for other EBPs, FleQ hydrolyses ATP to provide energy for remodelling of the closed sigma 54-RNA polymerase complex, allowing the expression of class II genes [59]. Expression of the class II genes generates membrane complex proteins and regulatory proteins, such as FlgM, and specific *Pseudomonas* regulatory proteins, such as FlhF, FleN (also called FlhG), and proteins of the two-component system (TCS) FleS/R (Figure 3B). The phosphorylated response regulator FleR is essential for class III gene expression in concert with sigma 54. The last promoter class, called class IV, is under the control of FliA [20].

Flagellar gene expression begins with the class I expression of *fleQ*, which is regulated by many different cellular processes, as *flhDC* in enteric bacteria. The activator FleQ acts in concert with the sigma 54 factor to drive the expression of genes through their class II promoters (see Table 1). These genes encode structural components of the membrane complex, motor, switch, flagellar export apparatus, and filament cap [20]. The FleP protein, expressed during flagellar class II gene expression, is specific to *Pseudomonas*. This protein is similar to FliT and influences the stability of the mature flagellar filament by an unknown mechanism [20,60]. Another gene, the *fliS’* gene, the name of which comes from its sequence similarity with *fliS*, is found only in a-type flagellin *P. aeruginosa* strains, but its function is still unknown [20]. Concerning FliL, in contrast to what has been described for *Salmonella* and *E. coli*, this cytoplasmic membrane protein is required for “swimming” motility in Pseudomonads [53].

In addition to the transcriptional hierarchy, the main differences in the flagellar assembly process between peritrichous flagellated bacteria and *Pseudomonas* species are the regulation of the location of the flagella and their number. In polar flagellated bacteria, such as *Vibrio* and *Pseudomonas*, the specific proteins FlhF and FleN (called FlhG in *Vibrio* and FleN or FlhG in *Pseudomonas*) are essential for generating a single polar flagellum [14]. FlhF protein is a signal recognition particle (SRP)-type guanosine triphosphate (GTP)ase and FlhG, a MinD/ParA-type ATPase. In *P. aeruginosa*, deletion of the *flhF* gene induces a reduction or loss of flagellar motility when cultured in 0.3% or 0.5% agar, respectively [61]. The authors proposed that FlhF enhances flagellar motor association. Moreover, deletion of the *flhF* gene in *P. putida* results in a random flagellar arrangement [62]. As FlhF is indispensable for flagellar polar localization, it is considered to be a polar landmark protein. In addition, FleN (syn. FlhG) is a negative regulator of flagellar biosynthesis and the deletion of *fleN* (*flhG*) results in multiple polar flagella [63]. In *P. aeruginosa*, FleN interacts physically with FleQ, the master regulator, to inhibit its transcriptional activity [16].

The detailed mechanism involved in FlhF/FleN regulation in *Pseudomonas* is still unclear. However, several studies have provided evidence on how the FleN-negative regulator controls FleQ activity. Dasgupta et al. showed that FleN physically interacts with FleQ through its ATP/GTP-binding site [16]. They assumed that the interaction of FleN with ATP induces a conformational change in FleN and permits FleQ binding. More recently, Baraquet et al. demonstrated that FleN decreases FleQ ATPase activity and, thus, its promoter-enhancing activity [59].

Hypotheses have recently emerged to describe this system in *Vibrio*. In *Vibrio*, *flhF* deletion led to either no flagella or, occasionally, cells with improperly localized flagella [63]. Kondo et al. showed that FlhF protein accumulation allows its binding to GTP and permits the formation of a homodimer [64]. This homodimer is in an “on-state” and acquires its polar placement. In general, SRP proteins (such as FlhF) bind to GTP to allow recognition of the amino-terminal domains of target nascent proteins and bring them to a membrane receptor [62]. Based on this general assumption, researchers suggested that FlhF may assist in MS-ring formation by recognizing FliF and bringing it to the membrane at the cell pole [14,62]. In parallel, the accumulation of FlhG first induces the repression of *flrA* (*Vibrio fleQ* homologue) expression and then its interaction with ATP, resulting in polar localization of FlhG-ATP, with the cooperation of HubP [64]. In the presence of FlhG, FlhF exerts its GTPase activity to generate GDP [64]. This GTP hydrolysis destabilizes the FlhF homodimers at the cell pole, which are then dispersed into the cytoplasm. Furthermore, the cytoplasmic dispersion of FlhF enhances FlhG ATPase activity. The binding of ADP to FlhG induces its dissociation from the membrane and results in the “off-state” of the protein [14]. In *Pseudomonas,* contrary to *Vibrio*, FlhF-GTP binding and GTP hydrolysis are not necessary for polar insertion of the flagella [65] and no homolog protein of HubP, indispensable for FlhG activity in *Vibrio*, has been found in *Pseudomonas*. Therefore, these observations are not directly transposable to *Pseudomonas.* However, in a recent study in *P. aeruginosa*, FlhF was shown to localize to the pole and to be able to interact with the C-ring protein FliG [66].

Furthermore, polar flagellated bacteria that contain the FlhF/FlhG regulatory system mostly possess TCSs, such as FleS/R in *P. aeruginosa*, FlgS/R in *C. jejuni*, or FlrB/C in *V. cholerae*. Such TCSs serve as an additional check point in flagellar biogenesis and are essential to allow rod and hook subunit expression, which are sigma 54 dependant, after correct assembly of the MS-ring/rotor/fT3SS complex [67]. Peritrichous bacteria do not possess a similar regulatory mechanism because rod and hook subunits are co-expressed with basal body components. In general, the sensor protein of a TCS recognizes a signal that induces its own phosphorylation. This phosphorylation is then transferred to the response regulator to modulate gene expression. In *P. aeruginosa*, FleS is a cytoplasmic protein that senses intracellular signals and transfers its phosphate to FleR. However, the interaction between FleS and FliG/M/fT3SS has not yet been shown, but Burham et al. demonstrated the importance of primary assembly in the activation of these TCSs [67]. Although this mechanism has not yet been elucidated in *Pseudomonas*, FlgS has been shown to physically interact with FliF and FliG in *Campylobacter jejuni* only if multimerized around the fT3SS. This interaction induces the phosphorylation of FlgS and activates its response regulator FlgR to induce the expression of rod and hook proteins. The activation of these TCSs results in the expression of the flagellar class III genes, which are sigma 54 dependent [20].

During class II and class III gene expression in *Pseudomonas*, the sigma factor FliA is sequestered by FlgM, which inhibits its activity [68]. As in *Salmonella*, the hook needs to reach a sufficient length to allow flagellar functionality. When the hook reaches the correct length, the anti-sigma factor is secreted to permit FliA activity and then class IV gene expression. However, this mechanism has not yet been totally elucidated.

The flagellar class IV genes include, notably, *fliC, fleL*, chemotaxis genes *che, mot* genes, and *flgM/N* [20]. The *fleL* gene is co-expressed with flagellin and is found only in monoflagellated bacteria genomes. The resulting protein is involved in the control of filament length, but the mechanism is still unknown [20]. In certain *Pseudomonas* strains, such as *P. fluorescens* F113, another flagellin gene, called *flaG*, belongs to the flagellar class IV genes and its deletion results in longer flagella [60].

The FliA regulon can differ between species. In contrast to *P. aeruginosa* PAK, the expression of *fliS* and *fliD* in *P. putida* KT2440 is dependent on FliA activity [69]. Another intriguing fact is that the *P. fluorescens* F113 strain is able to produce two types of flagella. As for other *Pseudomonas* strains, the first is regulated by FleQ, whereas the second is controlled by FlhD_4_C_2_, as in enterobacteria [70].

## 4. Fuelling of the Flagellar Machinery

### 4.1. Export of Flagellar Proteins

Assembly of the flagellar apparatus requires energy. It was first suggested that the T3SS export mechanism was based on the PMF because flagellar assembly was altered by the addition of uncoupler molecules [71]. The PMF is based on differences in membrane potential (Δψ) and pH (ΔpH) between the extracellular/intracellular media due to a chemical gradient and electron transport. This energy is used to export various flagellar components by a Sec-dependent pathway [71] and flagellar T3SS export [6]. The Sec-dependent pathway uses energy from ATP and the PMF [72] and is essential for export of the proteins involved in P-ring and L-ring formation. The FlhA protein of the fT3SS export gate possesses ion channel activity and is associated with an ATPase complex (FliI/J/H). The interaction between FlhA and FliJ ensures efficient utilization of the PMF for protein export [26].

The diameter of the fT3SS export channel is approximately 2 nm and requires that the substrate be unfolded to pass through such a narrow channel [27]. Many studies have proposed that the ATPase complex allows efficient substrate unfolding and recognition of the substrate/chaperone complex by the flagellar export gate [6]. Interestingly, flagellar assembly can occur in the absence of the ATPase FliI, albeit inefficiently but is favoured in the absence of FliH [71]. For example, in *Salmonella enterica*, the *fliI* deletion mutant shows lower “swarming” activity. However, deletion of the *fliH* gene in the Δ*fliI* strain increases “swarming” activity [73]. In this species, the authors proved that the PMF is the most essential energy source for flagellar export and that ATPases (InvC and Ssa) of non-flagellar T3SSs are not involved [73]. The ATPase complex is not required for fT3SS export in *Salmonella enterica* serovar Typhimurium. Mutations in *flhA* and *flhB* in the Δ*fliIH* mutant increase the probability of entry of flagellar proteins into the export gate, thus increasing export efficiency [74]. In the absence of the ATPase complex (FliHI), flagellar export requires both Δψ and ΔpH. This result suggests that ΔpH is required to overcome the absence of the ATPase, perhaps by providing energy for unfolding of the secretion substrates that are being funnelled into the export apparatus [75]. Briefly, FliH is required for efficient FliI_6_/FliJ complex formation. FliH can negatively regulate FliI ATPase activity, by forming FliH_2_/FliI complex, thus inhibiting FliI hexamerization [26]. The substrate-chaperone complexes are recognized by FliH_2_/FliI to permit their interaction with the ATPase complex and allow binding to FlhA. Kucera et al. proposed a model in which the ATPase complex allows the mechanical unfolding of substrate and the PMF permits its export. FliI_6_ appears to act as a stator, and FliJ is a coil-coiled filament that is able to move from a “ground state” to an “excited state”. The transition between these two states may explain the mechanical unfolding of substrates [27].

In addition, transport across the growing structure also requires energy. The mechanism that fuels the fT3SS is still unclear, but three models were proposed by Lee et al. to explain the exportation of late flagellar substrates [75]. Here, we will not discuss the last one because it was based on virulence-related T3SS. The first model corresponds to single-file diffusion driven by the energy generated by a difference in membrane potential [76]. This energy appears to be stored in the unfolded protein and then recovered when the secreted proteins refold to be incorporated into the nascent structure or upon exiting the secretion channel. The second model corresponds to a chain mechanism in which the flagellar subunits are pulled through the structure by an interaction between the carboxy-terminal domain of the secreted protein and the amino-terminal domain of the subsequently secreted one [25]. Here, folding of the secreted substrate upon exit from the secretion channel was proposed to provide the energy for pulling the monomer through the apparatus. This model is supported by the fact that flagellar monomers are able to interact with each other as they enter the flagellar channel. However, this explanation of how flagellar subunits are secreted through the nascent structure cannot explain the constant growth rate during flagellar assembly. It is still unclear how flagellar subunits are assembled at the tip of the flagella and how the monomer acquires the required energy to cross the nascent structure.

### 4.2. Generation of Flagellar Torque

The functionality of the extracellular flagellar appendage is based on its rotation due to the generation of torque. In *E. coli* and *Salmonella*, which possess peritrichous flagella, these structures have to all be bundled and rotate in the same direction to propel the bacteria in a straight line. Rotation of the flagellar filament is dependent on chemotaxis and energy generation.

#### 4.2.1. Chemotaxis

Chemotaxis is a system used by bacteria to sense extracellular molecules that drives the bacteria to move towards or away from them. Furthermore, there are various mechanisms involved in flagellar movement, including the switch between counterclockwise (CCW) to clockwise (CW) rotation, unidirectional rotation, no rotation, or variations in motor speed. The process of chemotaxis is well known in *E. coli* and *Salmonella*. These bacteria swim straight by rotating their filaments for a few seconds in the CCW direction. In this “run mode”, several flagellar filaments of a left-handed helical structure form a bundle and act as a screw. Alternatively, in the “tumble mode”, the bacteria tumble their bodies upon the CW rotation of the filaments for approximately 0.1 s. In this second mode, transition of the filament structure into a right-handed helix is induced. The bundle is then untangled, and the bacterium changes the direction of movement [23]. In addition, there are various chemoreceptors on the bacterial cytoplasmic membrane to induce the switch between CCW and CW rotation. These receptors are called MCPs (methyl-accepting chemotaxis proteins) and are essential for transducing the extracellular signal by their own methylation. Briefly, methylation of a MCP after binding to its specific chemoattractant inhibits activation of the two-component regulatory system, consisting of a sensor kinase, CheA, and a response regulator, CheY [6]. If bacteria move to a less favourable condition, fewer or no effector molecules bind to MCP, and CheA activity thus increases. This kinase phosphorylates the CheY proteins, which then interact with the flagellar motor protein FliM, causing a change in rotation of the rotor [6]. This leads to random reorientation of the bacterium, causing it to swim in a new direction.

In *Pseudomonas* species, chemotaxis can be viewed as an important prelude to ecological interactions, such as symbiosis, infection, or root colonization. For example, in many *P. fluorescens* strains, chemotactic responses to amino acids and organic acids are important for root-colonization [77,78,79]. Moreover, the chemotactic responses appear to be more sophisticated than those in enteric bacteria, which only possess one set of chemotaxis genes. For example, in the *P. aeruginosa* PAO1 strain, there are two different chemosensory signal transduction complexes (Che and Che2), which are localized at the cell poles. A review by Sampedro et al. summarized flagella-driven chemotaxis in Pseudomonads [80]. More than 26 MCPs are encoded in the genome of *P. aeruginosa* PAO1 strain to permit diverse chemotactic responses, of which six are only expressed during the stationary phase and allow remodelling between the two chemosensory complexes [81]. MCPs contain a periplasmic ligand-binding region (LBR) that recognizes chemoattractive or chemorepellent molecules and a cytosolic portion involved in the methylation/demethylation response [80]. Furthermore, MCPs can be divided into two classes, depending on their amino-acid chain length: Class I (150 aa) and Class II (250 aa). In *Pseudomonas*, class IIMCPs appear to possess two sites for the recognition of various ligands [80]. To summarize the chemotactic response in *Pseudomonas*, a decrease in the concentration of the attractant molecule induces methylation of the MCP. The phosphorylated CheA transfers its phosphate group to CheY. Based on sequence homology with *E. coli* CheY, in *Pseudomonas*, phosphorylated CheY is thought to interact with FliM. In contrast to *E. coli*, *Pseudomonas* produce only polar flagella, explaining the differences in the efficiency of their chemotactic mechanisms. Cai et al. proposed a new model in which Pseudomonads use a three-step swimming pattern of “run-reverse-pause” to change direction [82]. The duration of CW and CCW flagellar rotation is the same and increases when the bacteria follow a chemoattractant gradient. During the last step, the bacteria re-orient themselves by rotational diffusion.

#### 4.2.2. Energy Generation

The generation of flagellar torque requires ion motive force and depends on the flagellin filament arrangement (L-handed or R-handed) and the number of stator complexes surrounding the flagellum. For example, in *Salmonella*, the H^+^-driven motor spins at approximately 300 Hz, whereas in *Vibrio* spp. the Na^+^-driven motor is able to rotate at up to 1700 Hz [8]. It is important to note that certain bacteria can use both the sodium motive force (SMF) and PMF. For example, *V. parahaemolyticus* can produce two types of flagella: a constitutively synthesized, single, polar, sheathed flagellum driven by the SMF and hundreds to thousands of surface-induced, swarmer-cell-dependent, lateral flagella per cell driven by the PMF [7]. Briefly, to produce the energy required to produce flagellar torque, proton or sodium ions need to pass through the ion-channel of the MotA_4_/MotB_2_ stator complex. The MotB protein possesses an ion-conductive channel that contains an aspartate residue essential for cation binding (H^+^ or Na^+^) [83]. The protonation/deprotonation of MotB induces a conformational change in the cytoplasmic loop of MotA and permits C-ring movement by its own electrostatic interaction with the FliG protein. The flagellar motor driven by a single stator element shows 26 steps per revolution. Each step consists of at least two distinct processes: a torque-generation step, involving stator–rotor interactions, and a step for recovery of the original stator–rotor geometry [22]. In addition, electrostatic interactions between MotA and FliG are responsible not only for torque generation but also for stator assembly around the rotor [84]. Furthermore, stator complex formation is a dynamic mechanism in which MotB must interact with the peptidoglycan layer and MotA [24]. This stator complex is able to act as a mechanosensor to regulate stator incorporation around the motor in response to changes in external load or ion availability [22]. In more viscous environments, motors need to generate more force to rotate flagella and do so by recruiting stators to the motor in a load-dependent manner [22]. The number of stator complexes is upregulated in *E. coli* and other peritrichous bacteria when the external load increases. For example, in *E. coli*, it is possible to find up to 11 stator complexes around the flagellar structure to enable 11 rotation speeds [22]. In the *P. fluorescens* SBW25 strain, the flagellum appears to be more sophisticated than in peritrichous flagellated bacteria, permitting faster rotation of the flagellum [57].

In *E. coli* and *Salmonella*, deletion of the *motA* and/or *motB* genes results in the loss of motility. On the contrary, the deletion of these genes in the *P. aeruginosa* PAO1 strain did not result in the loss of motility [85]. A comparison of the genomes between the *E. coli* model and the *P. aeruginosa* PA14 strain highlights the presence of two sets of stator genes in *Pseudomonas*, called *motA/B* and *motC/D* based on their percentage of homology and appear to be present in all sequenced Pseudomonads [85]. The MotA/B and MotC/D stators are functionally redundant for swimming motility. However, the MotC/D complexes are essential for swarming motility at a high agar concentration. In addition to the MotA/B complex, movements of the *P. aeruginosa* PAK and PAO1 strains in viscous environments and over surfaces require the protein MotY [86].

## 5. Regulation of Flagella

The flagellar apparatus requires large amounts of energy for its production and function. Bacterial cells have to regulate these processes in concert with their metabolic states and coordinate the production and function of flagella with other cellular mechanisms. Consequently, flagella are regulated by multiple factors before and during the assembly process, as well as during movement. Each regulatory process can differ between bacterial species. Interestingly, correlations between flagella and physiological states can vary as a function of the bacterial species. For example, there are many differences in terms of the regulation of genes involved in flagella biosynthesis in *E. coli* and *Salmonella*. Bacteria of the *Salmonella* genus downregulate flagella-driven motility during conditions of starvation to save energy for survival. Conversely under the same conditions, flagella-driven motility is essential to *E. coli* to move to a more favourable environment and access nutrients. Thus, transcription of the corresponding genes is upregulated [48]. The main master regulators involved in these regulatory processes are FlhD_4_C_2_ or FleQ, but other flagellar components are also involved. For example, Mot proteins of flagellar stators also appear to play a part in exopolysaccharide production in *P. aeruginosa* [87]. A schematic network of the regulation of flagellar motility, focusing on bacteria from the *Pseudomonas* genus, is presented in Figure 4 to illustrate its complexity. In this review, we will focus on typical FleQ-dependent *Pseudomonas* flagella and will not detail the regulation of the second flagellar apparatus of *P. fluorescens* F113, which is under FlhD_4_C_2_-dependent regulation, similar to that in enterobacteria. For more information about the regulation of this second flagellar system, Barahona et al. have published a study [70].

### 5.1. Extrinsic Factors and General Mechanisms

As already mentioned, pH and proton concentration are essential for providing energy to the flagellar apparatus for its response to extrinsic factors. Minamino et al. showed that the intracellular proton concentration influences the speed of the flagellar motor by a reversible mechanism [88]. The authors proposed that the torque generator of the proton-driven motor has an intracellular proton-binding site where cytoplasmic protons kinetically interfere with rotation of the motor. In addition to flagellar torque generation, the PMF is known to influence the export of flagellar components [73]. However, other extrinsic factors, such as biochemical molecules or temperature, can also regulate the flagellar apparatus.

#### 5.1.1. Biochemical Molecules

Biochemical molecules regulate flagellar motility via chemotaxis, which allows bacteria to move towards certain nutrients or away from repellent molecules (Figure 4A). Most strains of *Pseudomonas* possess one or more sets of flagellar chemotaxis genes. Three sets of chemotaxis genes have been found in the *P. fluorescens* F113 strain, including genes encoding MCPs [89]. Each chemotaxis cluster is dedicated to a specific mode of action. For example, the CheA1 system is essential for chemotaxis under aerobic conditions, whereas CheA3 is necessary under conditions of denitrification [89]. The CheA2 system is required for optimal chemotaxis under both aerobic and anaerobic conditions. Thirteen of the 26 MCP-like proteins in the *P. aeruginosa* PAO1 strain have been functionally characterized. Ten MCPs mediate positive responses to oxygen (Aer and Aer-2), inorganic phosphate (CtpH and CtpL), malate (PA2652), amino acids and gamma aminobutyrate (GABA) (PctA/B/C), ethylene (TlpQ) and chloroethylenes and three mediate negative responses to chloroform and methylthiocyanate [80].

#### 5.1.2. Temperature

In addition to biochemical molecules, temperature is an important extrinsic factor that affects bacterial physiology, including motility governed by the flagellar apparatus. In *E. coli,* flagella are downregulated at high temperatures for several reasons [51]. First, an increase in temperature results in a decrease in the quantity of FlhD protein, explaining the small downexpression observed for flagellar class II genes. This decrease in FlhD protein is not explained by its degradation but probably by a decrease in translational efficiency. In addition, one protein of the flagellar export gate, FliP, appears to be unstable at 42 °C. Moreover, Rudenko et al. showed that FlgM secretion is inhibited at high temperatures, possibly explained by a decrease in hook-basal body assembly. The exact mechanism is still unknown, but the authors assume that such downregulation of flagella could occur during inflammation to escape detection by the immune system [51]. The regulation of flagella by temperature appears to be widespread in bacteria, including in *Pseudomonas*. Hockett et al. showed repression of the flagellar apparatus in *P. syringae* at 30 °C, which is a temperature slightly higher than that for optimal growth [3]. As in *E. coli*, the transcription of early flagellar genes (class I and II in *Pseudomonas* genus) is not affected by temperature. Furthermore, the authors showed that cytoplasmic accumulation of FlgM may be essential for such thermoregulation, but little is known about the mechanism of how this occurs (Figure 4B).

Environmental factors, such as high temperature and high osmolarity, are known to induce changes in DNA topology and the regulation of gene expression, affecting motility by intrinsic factors [1].

#### 5.1.3. DNA Topology

Certain proteins involved in DNA topology, such as histone-like proteins, can also be linked to the regulation of motility. For example, the histone-like protein H-NS in *E. coli* interacts directly with the flagellar motor protein FliG and is essential for flagella formation [90]. However, a recent study of Kim and Blair showed that H-NS/FliG interactions are not involved in the decrease of flagellar function [91]. Nevertheless, the authors showed that H-NS indirectly regulates flagellar synthesis. Moreover, H-NS-like proteins, such as VicH in *V. cholerae*, participate in the positive control of polar flagellar synthesis [1]. No similar mechanisms have yet been found in *Pseudomonas*, but other nucleoid-associated proteins may be involved in the control of motility.

#### 5.1.4. Membrane Stress

Membrane stress may also affect flagellar motility. For example, LPS truncation impairs flagellar gene expression in *E. coli* and *S. enterica* serovar Typhimurium [92]. Outer membrane perturbation by LPS modification enhances *rflP* (regulator of FlhDC proteolysis) gene expression. The RflP (syn. YdiV) protein binds to FlhD to drive proteolytic degradation of the FlhD_4_C_2_ master regulator by ClpXP, resulting in a decrease in motility [92]. In addition, RflP appears to be a sensor of the nutrient status of the cell [48,51]. In Pseudomonads, membrane stress does not appear to directly influence the flagellar apparatus [93].

### 5.2. Regulation by Two-Component Systems

TCSs also regulate flagella (Figure 4C). For example, the Gac (global activator) pathway is a TCS involved in the transduction of external signals to regulate numerous cellular pathways in *Pseudomonas* [94]. Briefly, GacS is an inner membrane sensor kinase and GacA the response regulator that permits the expression of *rsmX, Y,* and *Z* genes in small RNAs. In response to an extracellular signal, GacS is able to induce GacA autophosphorylation, which in turn leads to *rsmX, Y, Z* expression. The resulting small RNAs bind to RsmA, E, and I mRNA sequestering proteins, thus enabling target gene expression. In the *P. fluorescens* F113 strain, deletion of the *gacS* or *gacA* genes enhances swimming motility [95,96]. The repression of motility by the Gac system is due to the repression of *fleQ* gene expression, which inhibits FleQ-dependent flagellar gene transcription [96]. The Gac system also represses motility because RsmA sequestration by *rsmZ* RNA inhibits the repression of *algU* transcription. AlgU is an activator of *amrZ* expression [96], resulting in the synthesis of AmrZ protein, a transcriptional regulator that represses *fleQ* expression [97].

In *Pseudomonas*, the KinB system, another TCS, influences the formation of flagella. This TCS contains AlgB, a transcriptional activator and its cognate histidine kinase, KinB [98]. In the *P. aeruginosa* PA14 strain, KinB appears to be essential for the regulation of numerous virulence-associated phenotypes, such as quorum-sensing-associated virulence factors, biofilm formation, and motility [99]. Damron et al. proposed a model in which KinB is necessary for the degradation of MucA, an AlgU inhibitor, thus enhancing AlgU activity and, in turn, *amrZ* expression [98].

It is worth noting that the GacA/S signalling pathway in *Pseudomonas* is connected to another TCS regulatory system, the HptB pathway. The HptB pathway represses the GacA/S system when HptB is phosphorylated [100]. Briefly, inner membrane sensors allow HptB phosphorylation [101]. Phosphorylated HptB transfers its phosphate to HsbR and the activated HsbR-P dephosphorylates HsbA [100]. The non-phosphorylated HsbA interacts with FlgM. The HsbA-FlgM interaction enables the release FliA of into the cytoplasm, thus improving flagellar motility.

### 5.3. Regulation by Second Messengers

Flagella are also regulated by second messengers. A second messenger is a constituent of a pathway containing generally four components: two enzymes that produce or degrade the second messenger, an effector molecule that binds to the second messenger, and a target that produces a molecular output in response to the effector molecule [102].

#### 5.3.1. Adenosine 3′-5′ Cyclic Monophosphate (cAMP)

Adenosine 3′-5′ cyclic monophosphate (**cAMP**) is a second messenger produced by specific adenylate-cyclases in response to various signals and is degraded by a specific phosphodiesterase (PDE). This messenger interacts with a single effector, cAMP receptor protein (CRP), which binds to DNA regions to control target expression [102]. In *E. coli*, catabolic repression is involved in the regulation of flagella. Deletion of the *crp* gene in this species results in downexpression of *flhDC* and therefore the absence of flagella [90]. Catabolic activating protein (CAP), known to be the CRP in *E. coli*, binds to cAMP and appears to activate the expression of *flhDC,* which encodes the master regulator [1]. The cAMP-CAP protein and its consensus binding region are quite well conserved among several enterobacteria. Such sequence conservation suggests that cAMP-CAP activation may be a common mechanism to induce *flhDC* expression [1].

In *Pseudomonas*, Vfr is a homolog of *E. coli* CAP [103] and activates the expression of certain genes that are involved in virulence (*toxA*) or quorum sensing (*lasR*). Dasgupta et al. showed that Vfr is able to repress *fleQ* transcription by physically interacting with specific DNA regions [103] (Figure 4D). As Vfr is a CAP homolog, the authors hypothesized that cAMP could be a Vfr effector molecule that regulates its activity.

#### 5.3.2. Bis-(3′-5′)-Cyclic Dimeric Guanosine Monophosphate (c-di-GMP)

Bis-(3′-5′)-cyclic dimeric guanosine monophosphate (**c-di-GMP**) is considered to be a second messenger that modulates diverse cellular processes, such as virulence and motility. It is now recognized to play a central role in modulating the transition between planktonic and biofilm lifestyles in a large and growing number of bacterial species, including *P. aeruginosa*, *P. fluorescens*, *S. enterica, E. coli*, and *V. cholerae* [104]. The concentration of this second messenger depends on diguanylate cyclase (DGC) and phosphodiesterase (PDE) activity; c-di-GMP is produced by DGC from two molecules of GTP and is then degraded into 5′-phosphoguanylyl-(3′-5′)-guanosine (pGpG) by PDE [102]. The activity of these enzymes is modulated by various parameters to control the c-di-GMP concentration and thus bacterial physiology. This second messenger appears to be unique to bacteria and is generally able to bind proteins containing the specific c-di-GMP-binding domain (PilZ domain), such as FlgZ protein, with a 1:1 stoichiometry [105]. In general, a low concentration of c-di-GMP is associated with a planktonic lifestyle and therefore motility, in contrast to high concentrations, which are associated with a sessile lifestyle. Indeed, the link between flagella and biofilm formation is tight, but also finely regulated. For example, in *E. coli*, the overexpression of flagellin results in reduced adhesion [106], whereas motility promotes attachment [107]. Motility is regulated by c-di-GMP at different stages, from gene expression to post translational modifications [21].
First, c-di-GMP is able to act at the transcriptional level. For example, in *V. cholerae,* overexpression of CdgF, a DGC, induces a decrease in transcript levels of many class III and class IV genes and thus a decrease in “swimming” motility [21].Second, c-di-GMP regulates flagella at the post-transcriptional level. In the polar flagellated species *Caulobacter crescentus*, TipF protein appears to function as a PDE. Deletion of the *tipF* gene in this bacterial species affects motility but does not impair *fliC* or *flgE* transcription. Only the hook and flagellar filament are missing. In *E. coli*, MifA and MifB are two DGCs that promote c-di-GMP production. Both enzymes have been reported to act at a post-transcriptional level to decrease flagella production.Finally, c-di-GMP acts directly by altering flagellar function. The PilZ-like protein, YcgR, in *E. coli* or *S. enterica,* binds to c-di-GMP and consequently interferes with the association of Mot protein with FliG, which impairs flagellar rotation [21,22]. In *E. coli*, YhjH is a PDE and DgcE a DGC. These two proteins are essential for controlling c-di-GMP levels, and mediate flagellar activity [107]. This second messenger is recognized by two diguanylate receptors (DgrA/DgrB) that impair flagella function.

In Pseudomonads, c-di-GMP also acts at several levels (Figure 4E). For example, in the *P. fluorescens* F113 strain, FlgZ protein senses the c-di-GMP level and subsequently localizes to the cellular pole, where it can interact with the flagellar basal body, when a high concentration of c-di-GMP is reached [97]. Similarly, in *P. aeruginosa*, high c-di-GMP concentrations positively influence the polar localization of MotCD by interacting with FlgZ [22]. Furthermore, in *P. aeruginosa*, another protein, MapZ, appears to be involved in the regulation of flagella by c-di-GMP. At high c-di-GMP concentrations, MapZ interacts with the chemotaxis protein CheR1 and inhibits its methyltransferase activity and MCP methylation. Thus, a high c-di-GMP concentration is associated with a lower frequency of CCW/CW switching [108]. In addition, in *P. aeruginosa*, c-di-GMP is known to interact directly with FleQ, inhibiting its ATPase activity [59]. Thus, FleQ cannot activate flagellar gene expression at high c-di-GMP concentrations [109,110]. Many studies in *Pseudomonas* have highlighted the tightly linked regulation between flagella production and biofilm formation in relation with c-di-GMP levels. For example, GbcA is a conserved DGC among *Pseudomonas* species, and GbcA activity influences flagellar motility and biofilm production of *P. aeruginosa*, *P. fluorescens*, and *P. putida* [111]. In *P. putida*, MorA is a DGC [21]; deletion of the gene encoding MorA derepresses *fliC* expression. However, this phenotype appears to be limited to this species, as *fliC* transcription is not modified in a *P. aeruginosa morA* deletion mutant [106]. In addition, biofilm formation appears to be impaired for the *morA* deletion mutant in *P. aeruginosa* and *P. putida*.

Moreover, in *Pseudomonas*, two complex systems affect c-di-GMP production in response to contact with a surface [7]. The first system, called the Sad system, contains SadC (surface attachment defective mutant) and SadB proteins. In *P. aeruginosa,* SadC is an inner membrane-localized DGC that regulates c-di-GMP production and thus biofilm/flagella formation [112]. SadC catalyses c-di-GMP production in response to an unknown environmental signal. In *P. fluorescens* F113, the SadB protein participates in c-di-GMP turnover and « swimming motility ». Navazo et al. showed that the deletion of *sadB* in this strain reduced *fleQ* expression and flagella production [95]. Moreover, SadB appears to sense c-di-GMP levels and cooperates with SadC, which possesses sensor domains and diguanylate activity. SadB could thus be considered to be an effector of this system [95]. Furthermore, ethanol has been shown to mediate motility and biofilm production in *Pseudomonas* species through FlgZ protein in association with SadC [113]. Lewis et al. proposed a model in which ethanol induces a different ratio or occupancy of the MotAB and MotCD stators and boosts the interaction between the SadC DGC and GcbA. This interaction enhances their activities, increases the c-di-GMP concentration, and decreases flagellar rotation. Another system, called Wsp, is homologous to the flagellar chemotaxis system [114]. It consists of six proteins (WspA, C, D, E, F, and R), in which WspR works as a DGC when phosphorylated. WspA is a membrane-bound chemoreceptor that appears to be essential for surface sensing. WspR is able to phosphorylate itself and such phosphorylation is necessary for its oligomerization and thus its function [115]. In *P. fluorescens* F113, *wspR* expression impairs motility but not flagellin synthesis [95].

In *P. fluorescens* F113, BifA is involved in the regulation of « swimming » motility and biofilm formation [97]. Many studies have suggested that BifA is a PDE that counters SadC activity [7]. Recently, it was demonstrated in the *P. aeruginosa* PAK strain that overexpression of FcsR protein, a PDE, similarly prevents biofilm formation, flagellar motility, and chemotaxis [116].

### 5.4. Regulation by Quorum Sensing

Numerous bacterial behaviours are regulated in a cell density-dependant manner based on quorum sensing (QS). In Gram-negative bacteria, common QS systems involve acyl homoserine lactones (AHLs). The AHL QS system is able to regulate motility in numerous species of Gram-negative bacteria, but the final consequences can differ. For example, in *Sinorhizobium meliloti* or *Erwinia chrysanthemi* pv. *zeae*, QS represses flagellar expression, as QS-deficient mutants are hypermotile [117]. Conversely, in *E. coli*, QS is essential to produce the QseBC two component system, which is necessary for the activation of *flhDC* expression [118]. In *Yersinia enterocolitica* and *Burkholderia glumae*, AHL QS activates flagellar gene expression due to *flhDC* expression, as in *E. coli* [117,119]. At 37 °C, *Burkholderia glumae* produces polar flagella and deletion of the QS system leads to an aflagellated phenotype [119]. At 28 °C, the regulation of QS is different and could be essential for polar placement of the flagella [117].

Exogenous production of AHLs was recently shown to enhance flagellar-driven motility of *P. syringae* 11,528, but the exact mechanism is still unknown [120]. No direct regulation of flagella by QS has been yet reported for the model species *P. aeruginosa*. Many environmental *P. fluorescens* strains synthesize only small amounts of AHL or no common AHLs [121,122]. Similarly, *P. fluorescens* species do not produce PQS (2-heptyl-3-hydroxy-4-quinolone signal), a *P. aeruginosa*-specific QS signal [123]. However, these *P. fluorescens* strains are able to exhibit social behaviour (such as biofilm formation), which requires efficient QS. Gallique et al. proposed that the constitutively-expressed type six secretion system (T6SS) could serve as a signalling pathway for the *P. fluorescens* MFE01 strain, which lacks classical QS signal molecules [124,125].

### 5.5. Crosstalk between Flagella and the T6SS

The T6SS is one of the more recently discovered secretion systems [126]. It consists of a macromolecular machine that spans the bacterial cell envelope, similar to an inverted-bacteriophage [127,128,129]. T6SSs are composed of a membrane complex, a baseplate, and a tail. A membrane complex is essential for baseplate positioning, and the baseplate serves as a platform for contractile tail elongation. The contraction of the sheath propels effectors directly into target cells (eukaryotes and/or prokaryotes).

The same pathways regulate T6SS and flagella. Thus, crosstalk between these macromolecular machineries is probable. First, flagella and the T6SS are both regulated by temperature [3,130,131]. In addition, in *Pseudomonas*, T6SS and flagella are known to be regulated by TCSs (e.g., GacA/S) [132]. For example, many studies have demonstrated the influence of the GacA/S system on T6SS expression [95,133]. Allsopp et al. showed that AmrZ positively regulates H1 and H3-T6SS in *P. aeruginosa* and that RsmA represses all T6SS clusters [133]. In *P. aeruginosa*, bioinformatic analysis has suggested that two T6SS gene clusters (H2 and H3-T6SS) contain a sigma 54-binding site and appear to be under sigma 54 regulation [134]. Thus, as the expression of certain flagellar genes is controlled by sigma 54 factor, this factor is also considered to be a crosstalk protein that connects T6SS and flagella expression [132]. Moreover, as already discussed in this review, certain mutations in flagellar apparatus genes influence biofilm formation. Interestingly, biofilm formation is impaired for a number of T6SS-deletion mutants [124].

The virulence-related T3SS shares many similarities with fT3SS, suggesting cross-linked regulation between these two macromolecular systems. Indeed, fT3SS and the virulence-related T3SS are very often present at the same time and strongly modulate each other. For example, in *S. enterica* serovar Typhimurium, two secreted proteins of the virulence-related T3SS (SptP and SpoE) can be secreted through the fT3SS apparatus in the absence of their chaperone-binding domain [135]. Similarly, as the T6SS and virulence-associated T3SS are regulated by the same pathway [132,134], crosstalk between these two systems is likely. Indeed, recent studies have highlighted the link between the T6SS and flagellar-mediated motility. It was first shown in *E. coli* that deletion of a T6SS core-component gene (*icmF* syn. *tssM*) results in decreased motility [136]. In this strain, the motility defect was due to decreased *flhDC* and *fliA* expression. In *Citrobacter freundii*, deletion of the T6SS genes results in the downexpression of certain flagellar genes (*fliC*, *flgM, flgK, flgL, fliD, fliT*, and *motA)* and thus a decrease in flagellin export [137]. Another study revealed a link between T6SS and flagellar motility in *Acidovoras avenae* subsp. *avenae* RS-2 [138]. In this study, the authors highlighted a decrease in motility associated with the deletion of specific T6SS genes (*clpB syn. tssH*, *hcp syn. tssD*, *icmF syn. tssM,* and *pppA*). Moreover, in *Ralstonia solanacearum*, deletion of the *tssB* gene induces downexpression of the flagella regulon, affecting swimming motility [139]. Several studies have shown a correlation between T6SS and flagellar motility in *Pseudomonas* species. In the *P. aeruginosa* PAO1 strain, mutation of the *icmF3* gene resulted in impaired motility, without affecting expression of the flagella regulon [136]. Furthermore, deletion of the T6SS genes in the *P. fluorescens* MFE01 strain impaired flagellar motility [140]. This motility defect correlated with a decrease in FliA activity related to the cytoplasmic accumulation of FlgM [58] (Figure 4F).

## 6. Concluding Remarks

Flagella are complex nanomachines that are essential for bacteria and required for colonization, virulence, and recognition by host cells or the immune system. Flagella are also involved in biofilm formation and are important for the first step of colonization. This macromolecular complex is thus finely regulated in response to external stimuli, such as surface sensing or inflammation. Their production and function are under the control of a complex regulatory network in concert with the physiology of the bacteria. However, it is difficult to study this regulatory network in *Pseudomonas*, as certain details of the mechanisms of flagella production are still missing. For example, the transcriptional regulation of FliA is still unknown. In addition, mechanisms controlling the hook length, the number of flagella, and their polar insertion are yet to be elucidated. Furthermore, the exact function of several flagellar *Pseudomonas* proteins (i.e., FliS’, FliL, FleP, FlaG, and FleL) are still unknown. Studies are also hampered by differences between *Pseudomona*s species and strains in terms of regulatory processes. Studies on one of the *Pseudomonas* model strains may help to clarify these outstanding knowledge gaps. In this review, we considered the T6SS to be a new pathway that shares regulation with that of flagella, based on several studies. Moreover, it is clear that the regulation of flagella depends on cell-density, but the link between QS and flagella is still unclear in *Pseudomonas*. We believe that T6SS may be an alternative method to sense cell-density in certain *Pseudomona*s strains and then regulate flagella.

## Figures and Tables

**Figure 1 ijms-22-03337-f001:**
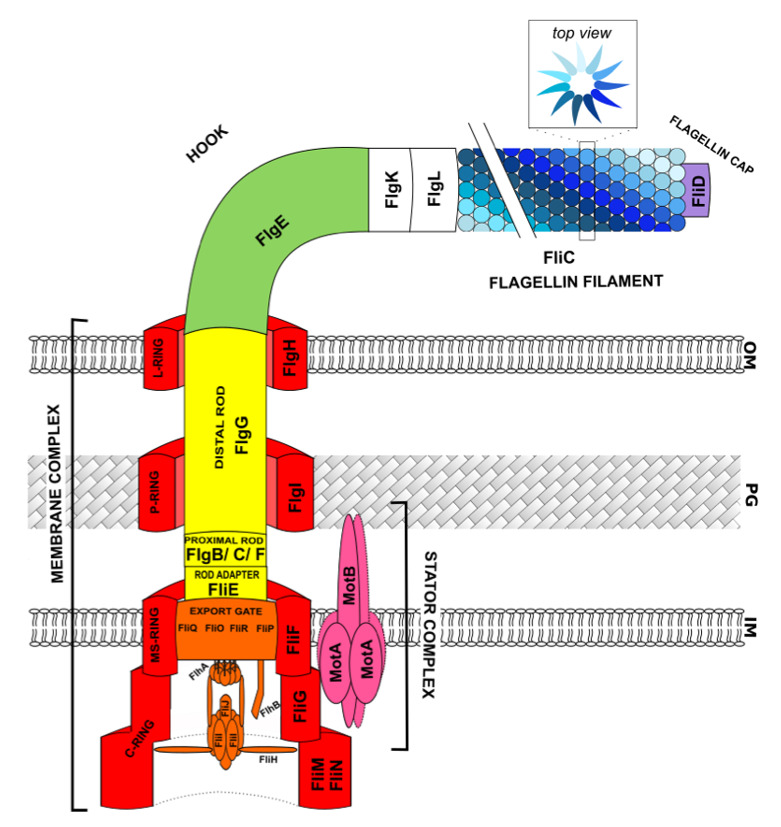
Schematic representation of the structure of the flagellar apparatus. IM, PG, and OM correspond to the inner membrane, peptidoglycan layer, and outer membrane, respectively. The flagellar membrane complex is composed of the basal body (in red), flagellar type 3 secretion system (in orange), and rod structure (in yellow). The flagellar filament (in various shades of blue) is connected to the hook (in green) by two junction proteins (in white). The flagellin cap (in purple) is the only cap protein that remains attached to the flagellar structure. The stator complex (in pink) is anchored to the peptidoglycan layer and interacts with the C-Ring to generate flagellar rotation.

**Figure 2 ijms-22-03337-f002:**
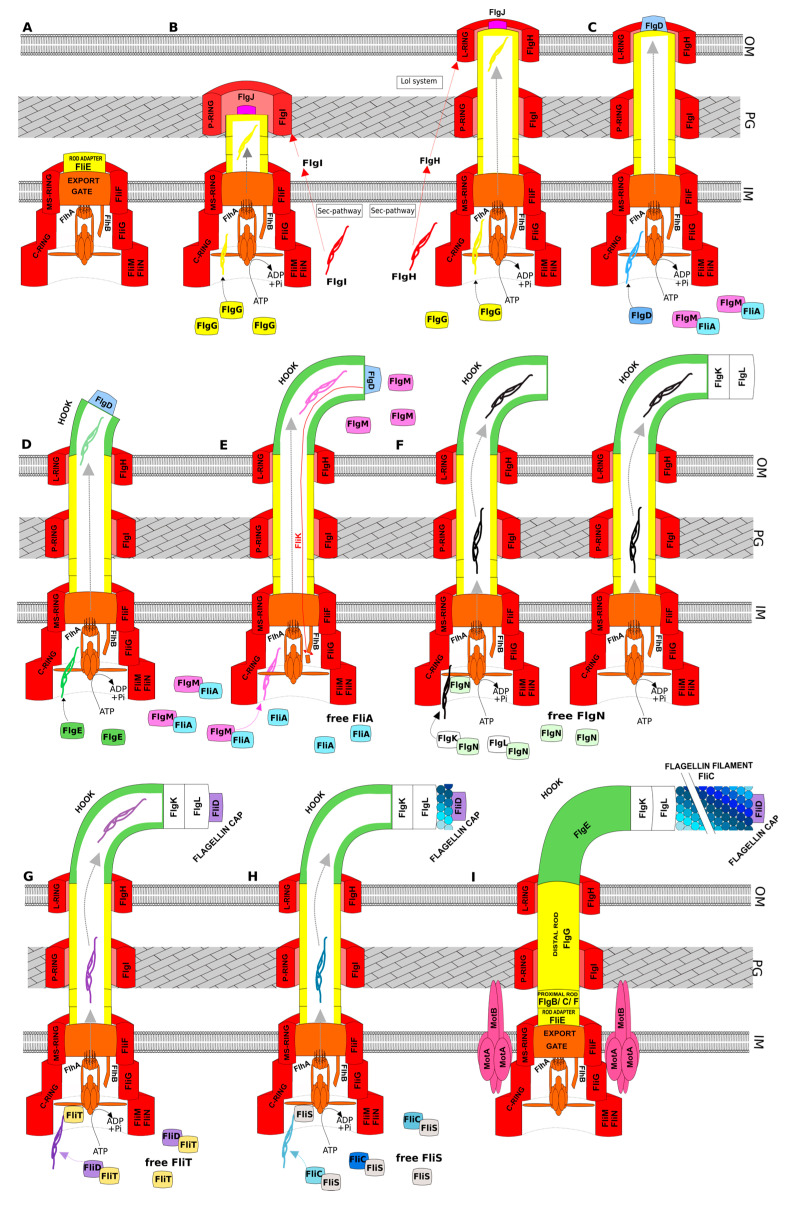
Schematic representation of flagellar assembly in nine arbitrary steps. IM: inner membrane, PG: peptidoglycan, OM: outer membrane. (**A**) Formation of the membrane complex and proximal rod. (**B**) Rod extension and P- and L-ring formation. (**C**) Hook cap protein insertion. (**D**) Hook extension. (**E**) Hook completion, substrate specificity switch, and FlgM secretion. (**F**) Junction protein addition. (**G**) Flagellin cap insertion. (**H**) Extension of the flagellin filament. (**I**) Insertion of the stator complex.

**Figure 3 ijms-22-03337-f003:**
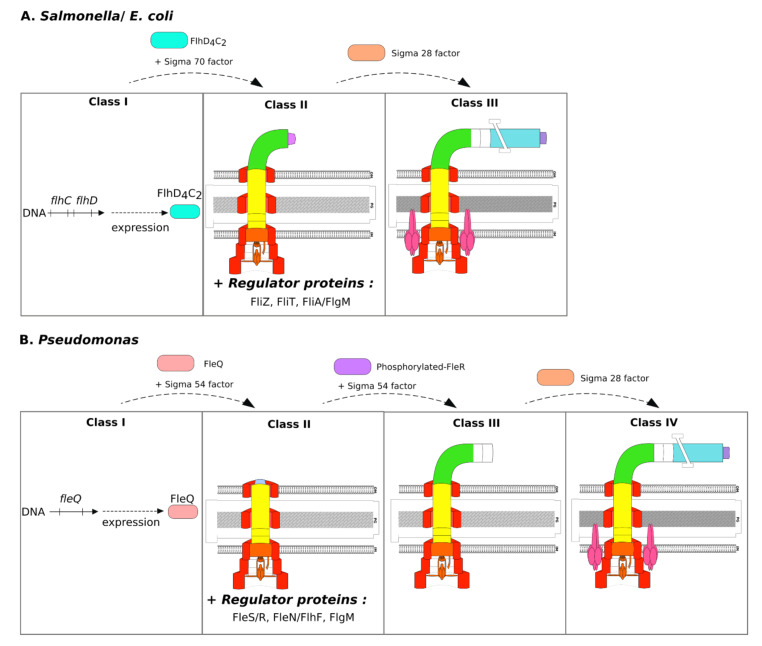
Schematic representation of the specifics of flagellar assembly in *Salmonella*/*E. coli* and *Pseudomonas*. Hierarchical transcription occurs in a three-step manner in *Salmonella* and *E. coli* (**A**) and in a four-step manner in *Pseudomonas* (**B**). Each box summarizes the production of a nascent flagellar structure and regulator proteins during expression of the corresponding class. Transcription is represented by the dotted arrows topped by the corresponding transcription factors.

**Figure 4 ijms-22-03337-f004:**
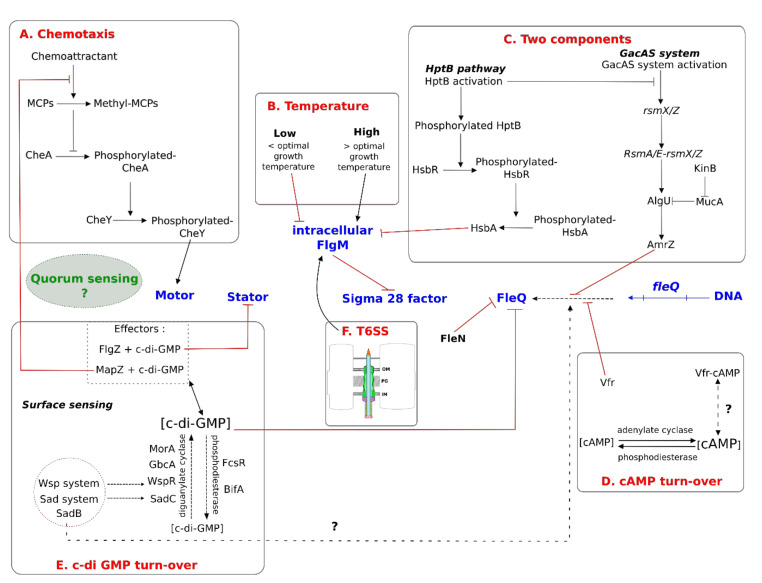
Regulation of flagella in *Pseudomonas*. The assembly and function of flagella in *Pseudomonas* are regulated by various mechanisms, such as (**A**) chemotaxis, (**B**) temperature, (**C**) two-component systems, (**D**) cAMP concentration, (**E**) c-di-GMP concentration, and (**F**) T6SS. Quorum sensing (QS) (represented in a green bubble) also appears to be involved but no data on the regulation of *Pseudomonas* flagella by QS are available. The key elements and regulators of flagella are written in blue. Black arrows indicate positive regulation, dotted arrows indicate potential positive regulation, double arrows indicate an interaction, dotted double arrows correspond to a potential interaction, blunt red lines represent negative regulation, and question marks indicate an unknown mechanism or interaction.

**Table 1 ijms-22-03337-t001:** Key genes involved in flagella.

*Salmonella/E. coli*		*Pseudomonas*		
Gene Name	Transcription Class ^1^	Gene Name	Transcription Class ^1^	Protein Activity
*che*	III	*che*	IV	Chemotaxis proteins
*flgA*	II and III	*flgA*	II	Flagellar basal body P-ring formation protein
*flgB, C*	II and III	*flgB, C*	III	Proximal rod proteins
*flgD*	II and III	*flgD*	III	Hook cap protein
*flgE*	II and III	*flgE*	III	Hook protein
*flgF*	II and III	*flgF*	III	Proximal rod protein
*flgG*	II and III	*flgG*	III	Distal rod protein
*flgH*	II and III	*flgH*	III	L-ring protein
*flgI*	II and III	*flgI*	III	P-ring protein
*flgJ*	II and III	*flgJ*	III	Distal rod cap protein
*flgK, L*	II and III	*flgK, L*	III	Hook/filament junction protein
*flgM*	II and III	*flgM*	II and IV	Anti-sigma 28 factor
*flgN*	II and III	*flgN*	II and IV	Hook/filament junction chaperone protein
*flhA, B*	II	*flhA, B*	II	Export gate proteins
*flhC, D*	I	*fleQ (syn. adnA)*	I	Master regulator, transcriptional activator
*fliA, sigma 28*	II and III	*fliA, sigma 28*	unknown	Sigma 28 factor
*fliC, fljB*	III	*fliC/flaA*	IV	Flagellin protein
*fliD*	II and III	*fliD*	II	Flagellin cap protein
*fliE*	II	*fliE*	II	Rod adaptor protein
*fliF*	II	*fliF*	II	MS-ring protein
*fliG*	II	*fliG*	II	C-ring protein
*fliH, I, J*	II	*fliH, I, J*	II	ATPase complex
*fliK*	II	*fliK*	III	Hook length control protein
*fliL*	II	*fliL*	II	Flagellum associated protein
*fliM, N*	II	*fliM, N*	II	C-Ring proteins
*fliO, P, Q, R*	II	*fliO, P, Q, R*	II	Export gate protein
*fliS*	II and III	*fliS, fliS’*	II	Flagellin chaperone protein
*fliT*	II and III	*fliT, fleP*	II and IV	Flagellin cap chaperone protein
*mot*	III	*mot*	IV	Flagellar motor protein
*ycgR*	III	*flgZ*	II and IV	c-di-GMP effector
*fliZ*	II and III			FlhD_4_C_2_ activator
*yhjH*	III			phosphodiesterase
		*flaG*	IV	Protein involved in filament length control
		*fleL*	IV	Protein involved in filament length control
		*fleR*	II	Two component system response regulator
		*fleS*	II	Two component system sensor protein
		*flhF*	II	Polar landmark protein
		*flhG, fleN*	II	FleQ anti-activator protein

^1^ Transcription class correspond to hierarchical transcriptions depending on different transcription factors. (For *Salmonella* and *E. coli* class I: Sigma 70; class II: FlhD_4_C_2_; class III: FliA. For *Pseudomonas* class I: Sigma 70; class II: Sigma 54 + FleQ; class III: Sigma 54 + FleR, class IV: FliA).

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
