# Peer review of "Pseudomonas* Flagella: Generalities and Specificities"

_ijms, 2021, doi:10.3390/ijms22073337_

Round 1
Reviewer 1 Report
The authors have written a very comprehensive and accessible review article covering most, if not all, facets of flagellar assembly. I have no suggestions or comments other than to thank the authors for their careful and authoritative treatment of this topic- well done!
One thought that the authors are free to ignore- is the mechanism by which FigE generates a bent section, as opposed to all the other Fig's that result in straight sections, known?
I did find a few minor spelling errors: 'Rod Adaptater' should be 'Rod Adapter' in Figure 1 and 'dependant' on line 311. This is why I must 'accept after minor revision'.
Again, this manuscript is a fantastic effort!
Author Response
The authors have written a very comprehensive and accessible review article covering most, if not all, facets of flagellar assembly. I have no suggestions or comments other than to thank the authors for their careful and authoritative treatment of this topic- well done!
Thank you !
One thought that the authors are free to ignore- is the mechanism by which FigE generates a bent section, as opposed to all the other Fig's that result in straight sections, known?
Thank you for this question. We added this information in lines 169 to 171 and added the corresponding reference (Fujii et al., 2018).
I did find a few minor spelling errors: 'Rod Adaptater' should be 'Rod Adapter' in Figure 1 and 'dependant' on line 311. This is why I must 'accept after minor revision'.
Rod adapter was modified in figure 1 and figure 2. Dependant was change by dependent.
Again, this manuscript is a fantastic effort!
Again, thank you !
Reviewer 2 Report
This is an extremely well written and thorough review on Pseudomonas flagella. The information is well researched and put together in a clear and concise manner. The figures are also very well done and add to the review.
Only two minor things are noted by this reviewer
Specific Comments:
Under Table 1 explain what transcription class I, II, III and IV mean
Lines 224-225: Is there a reason that H and I are not (H.) and (I.)? Any why they are also not bolded? Just a formatting issue
Author Response
This is an extremely well written and thorough review on Pseudomonas flagella. The information is well researched and put together in a clear and concise manner. The figures are also very well done and add to the review.
Thank you !
Only two minor things are noted by this reviewer
Specific Comments:
Under Table 1 explain what transcription class I, II, III and IV mean
These informations are added in a footer table 1
Lines 224-225: Is there a reason that H and I are not (H.) and (I.)? Any why they are also not bolded? Just a formatting issue
It was just a mistake !